# Proportion of children aged 9–59 months reached by the 2017 measles supplementary immunization activity among the children with or without history of measles vaccination in Lilongwe district, Malawi

**Hamilton Wales Kainga**  *, Steven Ssendagire, Jacquellyn Nambi Ssanyu, **Sarah Nabukeera, Noel Namuhani, Fred Wabwire Mangen**

Makerere University School of Public Health–Uganda, Kampala, Uganda

* kaingahamilton@gmail.com

## Abstract

### Background

The measles Supplementary Immunization Activity (SIA) was implemented in June, 2017 to close immunity gaps by providing an additional opportunity to vaccinate children aged between 9 months and up to 14 years in Lilongwe District, Malawi. This study was conducted to determine the proportion of eligible children that were reached by the 2017 measles SIA among those children with or without history of measles vaccination, and possible reasons for non-vaccination.

### Methods

A cross-sectional survey using mixed methods was conducted. Caretakers of children who were eligible for the 2017 measles SIA were sampled from 19 households from each of the 25 clusters (villages) that were randomly selected in Lilongwe District. A child was taken to have been vaccinated if the caretaker was able to explain when and where the child was vaccinated. Eight Key Informant Interviews (KIIs) were conducted with planners and health care workers who were involved in the implementation of the 2017 measles SIA. Modified Poisson regression was used to examine the association between non-vaccination and child, caretaker and household related factors. A thematic analysis of transcripts from KIIs was also conducted to explore health system factors associated with non-vaccination of eligible children in this study.

### Results

A total of 476 children and their caretakers were surveyed. The median age of the children was 52.0 months. Overall, 41.2% [95% CI 36.8–45.7] of the children included in the study were not vaccinated during the SIA. Only 59.6% of children with previous measles doses received SIA dose; while 77% of those without previous measles vaccination were reached

**Data Availability Statement:** All relevant data are within the manuscript and its Supporting Information files.

**Funding:** The author(s) received no specific funding for this work.

**Competing interests:** The authors have declared that no competing interests exist.

**Abbreviations:** CHAM, Christian Health Association of Malawi; EPI, Extended Program for Immunization; DHO, District Health Office; DHT, District Health Team; HSD, Health Sub District; LL, Lilongwe; MCV, Measles Containing Vaccine; MDG, Millennium Development Goals; MOH, Ministry of Health; MOV, Missed Opportunity for Vaccination; SDG, Sustainable Development Goals; SIA, Supplementary Immunization Activities; UNICEF, United Nations International Children's Emergency Fund; VPD, Vaccine Preventable Disease; WHO, World Health Organization.

by the SIA. Low birth order, vaccination history under routine services, low level of education among caretakers, unemployment of the household head, younger household head, provision of insufficient information by health authorities about the SIA were significantly associated with non-vaccination among eligible children during the 2017 measles SIA. Qualitative findings revealed strong beliefs against vaccinations, wrong perceptions about the SIA (from caretakers' perspectives), poor delivery of health education, logistical and human resource challenges as possible reasons for non-vaccination.

## Conclusion

Many children (41%) were left unvaccinated during the SIA and several factors were found to be associated with this finding. The Lilongwe District Health Team should endeavor to optimize routine immunization program; and community mobilization should be intensified as part of SIA activities.

## Background

Measles is a highly infectious and potentially fatal viral infection which continues to be a key contributor to child mortality particularly in sub-Saharan Africa and South Asia. While substantial progress has been made in recent years, measles still caused approximately 110,000 deaths globally in 2017 with most of the deaths occurring among children under the age of five years despite the availability of a safe and effective vaccine [1]. Infants and young children, especially those who are malnourished are at highest risk of dying. Immunization against measles directly contributes to the reduction of under-five child mortality, and hence the achievement of the sustainable development goal number 3 [2].

The World Health Organization (WHO) in its 2017 position recommends that countries should reach all eligible children with two doses of measles vaccine as the standard for all national immunization programmes [3]. In addition to the first routine dose of measles containing vaccine (MCV1), all countries should include a second routine dose of MCV (MCV2) in their national vaccination schedules regardless of the level of MCV1 coverage. Countries aiming at measles elimination should achieve ≥95% coverage with both doses equitably to all children in every district, and implementing high quality periodic campaign strategies referred to as SIAs [3]. SIAs are vaccination mass campaigns that are implemented in addition to routine vaccination programs with a recommended second dose opportunity to children of different ages regardless of their history of vaccination [4]. WHO also recommended that cessation of SIAs should be considered only when both MCV1 and MCV2 coverage of at least 90% had been achieved at national level for at least three consecutive years. For measles to be eliminated in at least five of the six WHO regions, including Sub-Saharan Africa by 2020 [5], countries should strengthen the routine vaccination program and address missed opportunities for measles vaccination in the routine immunization services through SIAs in order to achieve the necessary high levels of measles vaccination coverage required for population immunity [4].

Although SIAs are implemented widely, some populations that are not vaccinated through routine immunization services are often missed in such campaigns [6]. There is rich literature on why eligible children missed by routine immunization services are also left unvaccinated during mass campaigns. The systematic reviews on the impact of measles SIAs on reaching the zero-dose children that were missed by routine immunization services revealed household wealth, distance to the designated vaccination posts as reasons for non-vaccination [4]. Some

studies indicate lack of information about the SIA, parents' occupation, parents' level of education, age, parent's sickness, children's sickness, religious/cultural beliefs, and parents' belief that the disease is not serious as factors significantly associated with non-vaccination of eligible children [7, 8] Some studies have highlighted health system factors such as unavailability of vaccines, vaccinators not present at vaccination posts, and failure to health educate and mobilize communities [7–10].

In Malawi, MCV1 should be given when, or soon after, the child reaches 9 months of age. The second dose of measles containing virus (Measles 2) was introduced in Malawi in 2015 in the routine immunization program, which is given at 15 months of age [11]. All these are considered as valid doses of measles vaccine [12]. Routine immunization services are provided through static services (at health facilities) to every attending child who is eligible for measles immunization; and outreach services, which are offered at strategic community-based posts that are established at every 4–5 km away from each health facility, and in hard to reach areas. Immunization services are offered at least once in a month in all outreach posts [11] SIAs are organized periodically (usually every 3 years) to supplement the routine immunization services in an effort to interrupt the transmission and spread of diseases like measles. Despite the progress that has been made tremendously in reducing both mortality and morbidity associated with measles, Malawi has continued to report sporadic cases of the measles disease every 3 years [13]. The estimated coverage for measles vaccination across all districts in Malawi is 84%, but ranges from as low as 61% to high levels of almost 100% [14]. Measles vaccination coverage in Lilongwe District is suboptimal. According to the District Health Information System 2 (DHIS2) records, the measles vaccination coverage in the District has stagnated between 65% and 70% for the last three years. And the current measles vaccination coverage for the district as calculated from routine facility data is 67.4% [11, 15].

The last countrywide measles SIA was implemented in Malawi in June 2017. The SIA targeted all children who were aged between 9 months and up to 14 years irrespective of their vaccination status [11] In addition to the already existing outreach clinic sites, vaccination posts were created in selected churches, primary schools and football grounds in all the communities in every administrative division. Eligible children who were vaccinated were recorded in the health register. However, the vaccine was not being documented in the child's vaccination card, nor the child being given any document as an evidence of vaccination. The vaccination coverage attained following this SIA was not known because a post vaccination coverage survey was not conducted. Despite the successful implementation of the measles SIA in Lilongwe district in June 2017, there is little evidence to suggest that the measles SIA contributed to raising measles vaccination coverage because the coverage is still below 80%, and sporadic cases of measles still continue to be reported in the district. Therefore, this study was conducted to determine the proportion of eligible children that were reached by the 2017 measles SIA among those children with or without history of measles vaccination and possible reasons for non-vaccination.

## Materials and methods

### Study setting

The study was conducted in Lilongwe District. Lilongwe is the capital city of Malawi with an estimated population of 1,077,116. [16]. The city is located in the central region of Malawi, near the borders with Mozambique and Zambia, and it is an important economic and transportation hub for central Malawi. It was named after Lilongwe River. Administratively, there are 22 constituencies, and over 500 neighborhoods or villages. Malawi people are of Bantu origin, and Lilongwe comprises mainly the Chewa ethnic group whose main occupation is farming and trade.

In the Malawi health system, health services are predominantly delivered by the public sector (free at the point of use), Christian Health Association of Malawi (CHAM–which is an umbrella body of Christian faith-based health facilities operating on a not-for-profit basis); the private health sector (which charges user fees); and the Non-Governmental Organization (NGO) sector [12]. Public and CHAM health facilities constitute the two largest providers, collectively providing about 90% of health services in urban and rural areas [17]. Lilongwe City is served by 47 public health facilities, of which 11 belong to church missionaries and 36 owned by the government. There is a district hospital, and a central hospital which serves as a specialized facility for the central region of Malawi. In addition, there are over 100 private health facilities [15].However, accessing health care in all levels of the health system still remains a challenge for marginalized children living in the rural areas, urban slums and outskirts of the city [10].

## Target population

The study targeted children who were aged 9–59 months during the implementation of the June 2017 measles SIA in Lilongwe District, Malawi. These children were 31–81 months of age in April, 2019 when the survey was conducted. The primary sampling units were households. The study respondents were caretakers or mothers (of at least 15 years of age) of children who were eligible for 2017 measles SIA.

## Study design

A cross sectional survey using mixed methods was conducted. Quantitative data were collected from caretakers through a household survey. Key informant interviews (KIIs) with health care workers were used to obtain qualitative data.

## Sampling procedure

A two stage sampling technique was employed for the selection of participants for the household survey. Administratively, there are 528 villages in Lilongwe district. These villages represented clusters; therefore, 25 villages (clusters) were randomly selected using probability proportionate to size (PPS). In each selected village, 19 eligible households were selected. Because the total number of households in all the villages was unknown, it was difficult to select the households randomly. The center of the village, or any feature such as a church or market, was located using a local guide. The first eligible household in the village was purposively selected. Then every nearest eligible household was systematically visited.

In each household, a primary caretaker was identified who was interviewed using the interviewer administered electronic questionnaire to obtain information on social demographics of the child, the caretaker, and the entire household. Information was sought from one eligible child in household. If there were more than one eligible child in a household, information was sought from only one child who was randomly selected in that particular household. We relied on maternal recall of measles vaccination status of a child because the measles SIA vaccine was not documented in the child's vaccination card. A child was taken to have been vaccinated if the caretaker was able to explain when and where the child was vaccinated.

The minimum sample size for this study was determined according to Kish Lesley's formula [18] with the following assumptions: the percentage point for α error = 5%, precision δ taken as 5% and we estimated that 16.9% of eligible children would be unvaccinated during the mass vaccination campaign according to [4].

We planned to conduct twelve KIIs with health care workers who were involved in the planning and implementation of the vaccination campaign. However, only eight KIIs were conducted because the saturation point was attained by the seventh interview.

### Eligibility criteria

Caretaker-child pairs were eligible for study inclusion if they were from households with a child who was aged 9 months to 59 months during the June 2017 measles SIA, and have given consent to participate in the study.

### Research team

The research team comprised the principal investigator (PI) and eight research assistants. The assistants had a minimum of post-secondary school education with experience in data collection. They were not related to the study participants, nor to the principal investigator. They were trained for two days before the start of data collection exercise so that they became familiar with the statement of the problem, objectives of the study, sampling procedure, data collection tools and plan for data collection and interview techniques. The PI was responsible for collection of qualitative data by facilitating and conducting KIIs.

### Data collection and measurements

Both qualitative and quantitative data were collected simultaneously. Child's history of vaccination was determined according to the child's vaccination card and/or caregiver self-reports of prior vaccination. Specifically, for routine vaccination, the interviewer asked to see the child's vaccination card if it was available and noted the date of vaccination recorded on the card. If the card was not available, interviewers would ask the caretaker/guardian if the child had ever received measles vaccine at the age of 9 months or older, and the number of measles vaccine doses a child received at a health facility. In addition, caretakers were asked whether their children participated in the 2017 measles SIA (with possible answers being "yes" or "no"). Since the SIA vaccine was not being documented in the child's vaccination card, nor the child being given any evidence of vaccination, a child was taken to have been vaccinated during the measles SIA if the caretaker was able to explain when and where the child was vaccinated in that community. The WHO and some authors argue that recall by vaccination card is considered the best practice for determining vaccination coverage in a household survey and is preferable over self-reported recall [19]. However, some previous studies also concluded that maternal self-reports are trustworthy and are as good as vaccination cards [20, 21]. In addition, data on socio-demographics of the mother and child were collected using an electronic interviewer administered questionnaire.

We planned to conduct twelve KIIs with health care workers from public health facilities who were involved in the planning and implementation of the vaccination campaign. These were environmental health officers (health sub district supervisors) who were responsible for planning, and nurses who were responsible for the implementation. However, only eight KIIS were conducted because the data saturation point was reached after the seventh and eighth interviews. These were purposively selected from three health sub-districts, namely, Bwaila, Nanthenje and Mitundu. A semi structured KII guide was used for each respondent. A priori themes included in the KII guide were about the reasons for non-vaccination of eligible children emanating from health information system, human resources, financing and logistical challenges. Two KIIs were conducted with SIA planners (Environmental Health Officers) and six with the nurses who were involved in the SIA implementation. We used audio recording to collect the data. Field notes were made during and after the interview. Each interview lasted between 15 and 20 minutes, and were all in English. All the KIIs were carried out by the principal investigator who did not have any relationship or prior knowledge with the key informants. The outcome variable in this study was measles vaccination during 2017 measles SIA to an eligible child who had no contraindication to vaccination. This was a binary outcome ('yes' or 'no', with 'no' representing non-vaccination).

### Pre-testing and field editing of data

The data collection tools were pre-tested in five households that were not part of the sample. Thereafter the tools were adjusted for content validity before being used in the field. Filled electronic questionnaires were checked while still in the field for completeness and those found incomplete and erroneous were corrected before the respondents were discharged.

### Data management

**Data entry and cleaning.** The interviewer-administered pre-coded electronic questionnaire was used both for collecting data from eligible households and entering the data simultaneously using Open Data Kit (ODK) installed on mobile phones. The data were cleaned and edited when imported into excel spreadsheet. The data were coded and then checked for consistency. Explorative data analysis (EDA) was carried out to check for missing values and completeness of data for all variables of interest. The data were then exported to STATA 14 for further cleaning, manipulation and analysis.

**Data analysis.** Data were analyzed using STATA version 14 (StataCorp LP, TX, USA). Univariate analysis was done to summarize the data on respondent characteristics utilizing tables and graphs. Means with standard deviations were used to summarize normally distributed continuous variables while medians with interquartile ranges were used for continuous variables that remained non-normally distributed even after transformation, and percentages for the categorical variables.

**Bivariate analysis.** Modified Poisson regression analysis was carried out to estimate associations between missed opportunities for measles vaccination and the risk factors. The measure of association used was the prevalence ratio (PR).and the corresponding 95% confidence intervals. The effect of each independent variable on the dependent variable was checked at a significance level of 0.05.

**Multivariable analysis.** Variables with p-value < 0.2 from bivariate analysis were included in the final multivariable model. Log likelihood and Akaike's Information Criteria (AIC) were used to determine the goodness of fit of the adjusted final model in comparison to the preceding models. The AIC value for each subsequent model was compared, and the model with the lowest value was considered to be the best fit model [22]. The presence of multicollinearity was checked among independent variables using Variance Inflation Factor (VIF) at a cutoff point 10. Predictors having a VIF value less than 10 indicated the absence of multicollinearity [23].

**Qualitative data analysis.** The qualitative data were analyzed manually by the principal investigator. Verbatim transcription was done to generate data from each KII. Both Deductive and Inductive approaches were used to analyze the data. However, the analysis was more deductively done as the principal investigator had prior themes in the KII guide. After transcription of the audio data, the material was systematically read through in order to identify codes, categories, and themes. During analysis, new categories were developed inductively. The underlying meaning of the categories was formulated into a theme. Illustrative quotations were selected. Qualitative data analysis was done after quantitative data analysis to identify health systems related factors that were classified according to the health system building blocks framework [24]. Fig 1 below summarizes the quantitative and qualitative methods used in this study.

### Ethical approval

Approval to conduct this study was obtained from Makerere University School of Public Health Higher Degrees Research and Ethics Committee. Permission was also sought from the

**Quantitative data collection & analysis**          **Qualitative data collection & analysis**

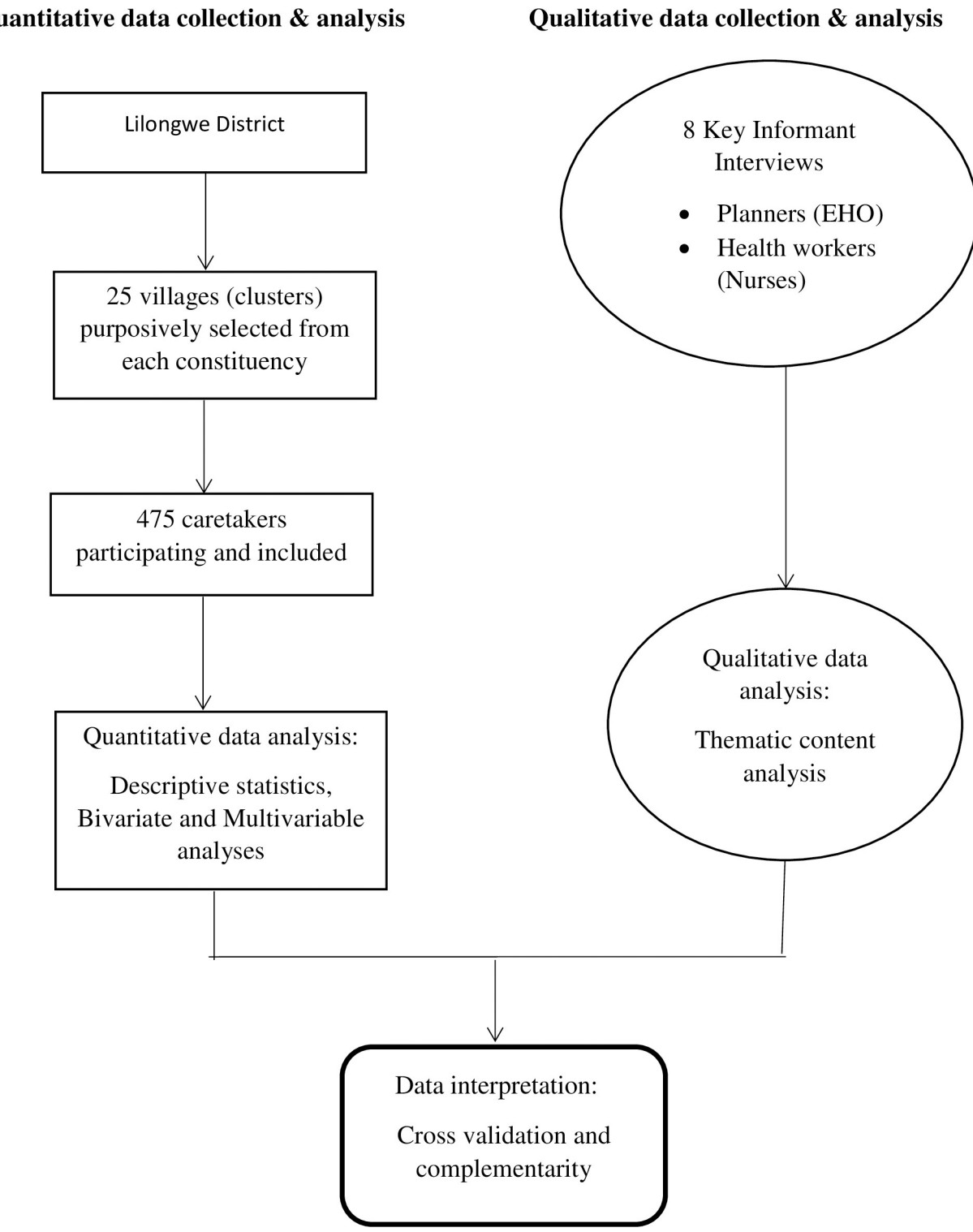

**Fig 1. Summarizing the mixed methods used in this study.**

EPI of the Lilongwe District Health Office and the National Health Sciences Research Committee of Malawi. Respondents were read an informed consent which clearly stated the

following 1) the purpose of the study, 2) what participation in the study would involve, 3) how confidentiality and anonymity would be maintained, 4) the right to refuse to participate in the study or to withdraw from the study without any penalty, 5) the benefits of participating in the study. Confidentiality and anonymity were maintained by the use of code numbers in the interviews during data collection. After explaining the study, the participants who could read and write were asked to sign a consent form.

## Results

Four hundred and seventy six eligible children were included in the study and 52.3% (249) were females. Their ages ranged from 31 months to 81 months with a median age of 52.0 months. Almost all respondents were mothers of these eligible children (97.9%, 466/476). Their mean age was 29.7 years (SD 6.9 years). Most of them [85.7% (408)] were married. More than half of the respondents (61.3%) had primary school education. Lower proportions of those included (16.0%) and (2.1%) had secondary school and tertiary education respectively. Only 16% were fully employed. Table 1 shows background characteristics of the study participants.

**Table 1. Background characteristics of the study participants.**

| Characteristic | Number | Percentage |
|---|---|---|
| Sex of the child | | |
| Male | 227 | 47.7 |
| Female | 249 | 52.3 |
| Child's age (months) in 2017 | | |
| 9–21 | 195 | 41.0 |
| 22–34 | 117 | 24.6 |
| 35–47 | 80 | 16.8 |
| 48–59 | 84 | 17.7 |
| Birth order | | |
| 1st born | 129 | 27.1 |
| 2nd - 4th born | 273 | 57.3 |
| 5th + | 74 | 15.6 |
| Place of delivery | | |
| Home | 14 | 2.9 |
| Health facility | 462 | 97.1 |
| Caretaker's age (years) | | |
| 16–24 | 120 | 25.2 |
| 25–34 | 232 | 48.7 |
| 35+ | 124 | 26.1 |
| Marital status | | |
| Currently Married | 408 | 85.7 |
| Currently not married | 68 | 14.3 |
| Caretaker's education level | | |
| No education | 108 | 22.7 |
| Primary | 283 | 59.5 |
| Secondary | 76 | 16.0 |
| Post-secondary | 9 | 1.8 |
| Caretaker's employment status | | |
| Unemployed | 400 | 84.0 |
| Self employed | 41 | 8.6 |
| Employed | 35 | 7.4 |

### Proportion reached by the SIA among children with or without history of vaccination

Seventy six percent, 76% [362/476, 95% CI: 73.4–79.0] that participated in the study were vaccinated against measles at the age of 9 months or older under routine immunization services. Overall, 41.2% [95% CI 36.8–45.7] of these 476 children missed the opportunity of receiving the measles vaccine during the 2017 measles SIA. Only 59.6% [95% CI 54.9–64.1] of those that received measles vaccine under routine immunization services participated in the SIA. On the other hand, 51.5% [95% CI 34.7–68.0] of eligible children that did not receive the measles vaccine at the clinic under routine services also missed the opportunity to get vaccinated during the measles SIA. Figs 2 and 3 below respectively show the overall proportion of eligible children that were not vaccinated, and the proportion of zero-dose (whithout history of measles vaccination) children who were also missed during the 2017 measles SIA in Lilongwe District.

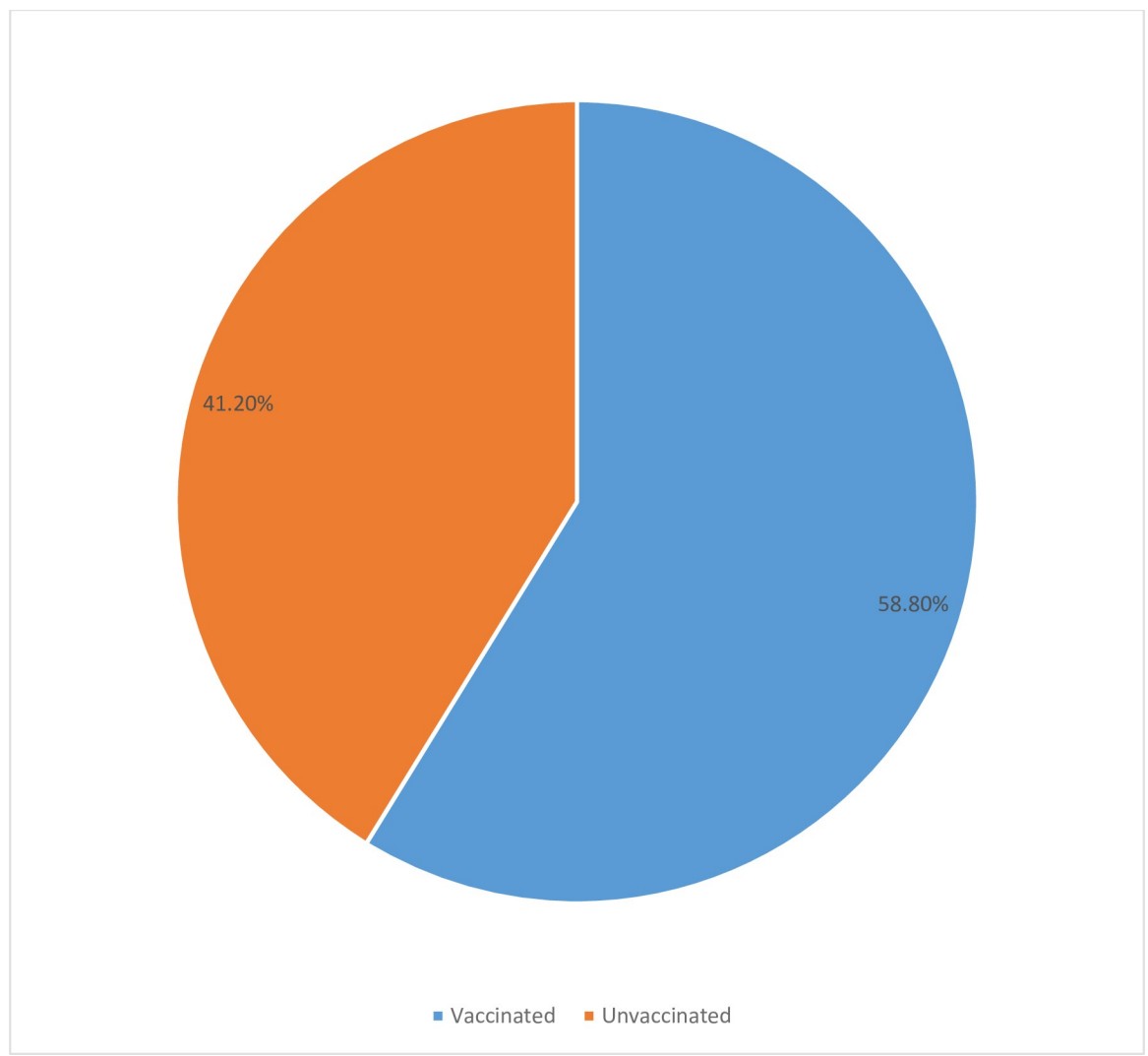

**Fig 2. Showing the overall proportion of eligible children that were left unvaccinated during the SIA in Lilongwe.**

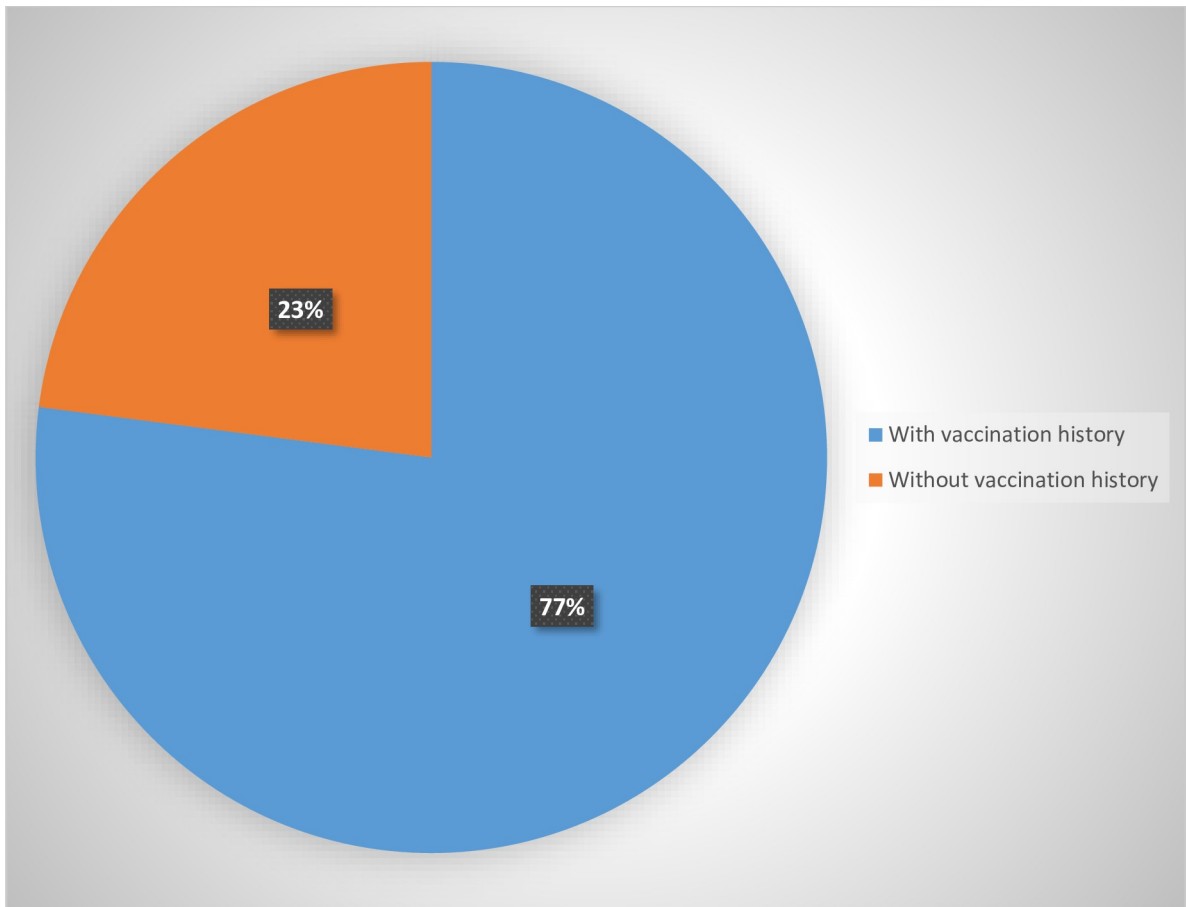

**Fig 3. Showing the proportion of eligible without history of measles vaccination who were also missed during the SIA.**

### Factors associated with non-vaccination and possible reasons for non-vaccination

Socio-demographic characteristics such as maternal age, education level, employment status, marital status, child's age, child's sex, place of delivery, birth order, previous history of vaccination, number of under-five children in household, media exposure, religion, sex of household head, age of household head, education level of household head, and employment status of household head were studied to identify potential factors associated with measles non-vaccination during the mass vaccination campaign. Table 2 below shows bivariate and multivariate analysis of factors associated with non-vaccination during the measles SIA.

After adjusting for other factors, the prevalence of non-vaccination among children with higher birth order was 0.80 [APR = 0.80, 95% CI 0.65–0.98] times that of first-born children. In addition, the prevalence of non-vaccination among children with history of measles vaccination under routine programs was 35% [APR = 0.65, 95% CI 0.47–0.88] lower than that of zero-dose children (i.e. with no history of vaccination) during the 2017 measles SIA. The prevalence of non-vaccination among children born from mothers or caretakers having at least primary school education was 23% [APR = 0.77, 95% CI 0.62–0.96] less than that of children born from mothers with no formal education holding other factors constant. The prevalence of non-vaccination was 28% greater among children whose caretakers were unemployed than that of their counterparts from employed caretakers [APR = 1.28, 95%CI 1.12–1.42]. The

**Table 2. Bivariate and multivariable analysis of factors associated with non-vaccination.**

| Factor | Prevalence | Unadjusted PR(95% CI) | Adjusted PR (95% CI) |
|---|---|---|---|
| Birth order | | | |
| 1st born | 50.8 [42.1–59.4] | 1.0 | 1.0 |
| 2nd - 4th born | **38.4 [32.8–44.3]** | 0.75 [0.55 1.01] | **0.80 [0.65–0.98]** * |
| 5th + born | **34.7 [24.6–46.4]** | **0.60 [0.44–0.82]** | 0.73 [0.53–1.02] |
| Caretaker's age | | | |
| 16–24 | 50.0 [41.1–58.9] | 1.0 | |
| 25–34 | **37.5 [31.5–43.9]** | **0.75 [0.59–0.96]** | |
| 35+ | 39.5 [31.3–48.4] | 0.79 [0.60–1.05] | |
| Caretaker's education | | | |
| No education | 50.1 [40.2–59.8] | 1.0 | 1.0 |
| Primary education | **38.0 [32.6–43.7]** | **0.76 [0.59–0.97]** | **0.77 [0.62–0.96]** * |
| Secondary educ | 40.7 [30.3–52.2] | 0.82 [0.58–1.14] | 0.93 [0.67–1.28] |
| Post-sec education | 48.0 [24.4–73.6] | 1.00 [0.52–1.92] | 1.55 [0.90–2.68] |
| Employment status | | | |
| Employed | **25.0 [12.2–40.4]** | **1.0** | **1.0** |
| Self employed | 41.3 [27.9–56.0] | 1.12 [0.95–1.33] | 1.19 [0.89–1.27] |
| Unemployed | 51.6 [46.2–59.4] | 1.48 [1.31–1.64] | 1.28 [1.16–1.40] |
| Age of household head | | | |
| 16–24 | 59.0 [42.9–73.3] | 1.0 | |
| 25–34 | **44.2 [37.5–51.1]** | 0.75 [0.55 1.01] | 0.79 [0.55–1.14] |
| 35+ | **35.5 [29.6–41.9]** | **0.60 [0.44–0.82]** | **0.63 [0.40–0.99]** * |
| Employment of h/h head | | | |
| Employed | 47.6 [40.4–54.8] | 1.0 | 1.0 |
| Self employed | 41.4 [34.2–49.0] | 0.87 [0.69–1.10] | 0.84 [0.67–1.04] |
| Employed | **28.8 [20.9–38.3]** | **0.61 [0.43–0.85]** | 0.69 [0.51–0.95] * |
| Providing info about SIA | | | |
| No | 85.2 [77.1–90.8] | **1.0** | 1.0 |
| Yes | **28.3 [23.8–33.1]** | **0.33 [0.28–0.40]** | 0.49 [0.37–0.65] ** |
| History of vacc | | | |
| No | 77.3 [66.1–93.0] | 1.0 | 1.0 |
| Yes | **33.6 [21.6–44.7]** | **0.67 [0.55–0.82]** | **0.65 [0.47–0.88]** ** |

Key

* P-value < 0.05

** P-value < 0.001; APR: Adjusted prevalence ratio.

prevalence of non-vaccination among eligible children decreased with increasing caretaker's age but this relationship disappeared in the adjusted model.

Similarly, after controlling for other variables, the prevalence of non-vaccination during the 2017 measles SIA was 37% [APR = 0.63, 95% CI 0.40–0.99] lower among children coming from households where the household head was at least 35 years old than that among their counterparts having a household head aged 16–24 years. The prevalence was 31% [APR = 0.69, 95% CI 0.51–0.95] lower if the household head was employed "Health authorities providing information about the SIA" was much significantly associated with non-vaccination among eligible children. The prevalence of non-vaccination among children from households that indicated that health authorities provided information about the measles SIA was 0.66 [APR = 0.66, 95% CI 0.50–0.87] times that of children from households household that

indicated they were not provided with information about the measles SIA by the health authoritis. Similarly, "getting information about measles vaccine and its safety" was highly significantly associated with vaccination of eligible children against measles during the campaign. The prevalence of non-vaccination was 51% lower [APR = 0.49, 95% CI 0.37–0.65]. This was also highlighted in the KIIs where it was revealed that some parents had a wrong perception and wrong understanding about the 2017 measles mass vaccination campaign regarding the eligiblity and safety of their children because information was not communicated in a proper way.

*"The problem was that some parents believe that their children who are under two years of age receive vaccines at health facilities or under-five clinics. So because of that perception, they did not bring such children for vaccination. I think may be the messages about the campaign were not well clarified."* (Nurse from Bwaila HSD)

Discussions with the planners revealed that the mass vaccination campaign was also hampered by flaws existing in the health information system. This was manifested by the use of wrong estimates of health facility-catchment areas during planning and implementation of the SIA. This ultimately contributed to the logistical and human resources challenges in some health zones.

*"Usually we use National Statistical Office (NSO) population figures of all health sub districts for planning purposes. So it happened that in some areas we came up with wrong population estimates because of following NSO figures. This led to inadequate supplies being allocated, and few health workers deployed to such health zones."* (EHO from Bwaila HSD)

A common emerging theme concerned the duration of the measles SIA. All the informants complained that the five days of implementing the campaign was very little considering that they targeted all eligible children, who were representing 46% of the country's total population (children of 9 months up to 15 years).

*"The period (five days) of implementing the campaign was very little to vaccinate all eligible children. May be some children who were missed in the first days because their parents were not available, or because of funerals in the villages would have had another opportunity to be vaccinated if government and other partners would consider extending the period of the campaign may be up to ten days."*(Nurse, Mitundu HSD)

## Discussion

A cross-sectional mixed methods study was conducted in Lilongwe district, Malawi, to determine the proportion of eligible children that were reached by the 2017 measles SIA among those children with or without history of measles vaccination in Lilongwe district, Malawi, and to explore the reasons for non-vaccination. Overall, 41.2% of eligible children included in the study were not reached by the SIA during the campaign. Only 59.6% of those children that received measles vaccine under routine immunization services participated in the measles SIA. On the other hand, 51.5% of eligible children that did not receive the measles vaccine at the clinic under routine services also missed the opportunity to get vaccinated during the measles SIA. The possible explanation for this finding might be that there was not enough sensitization and mobilization on the part of planners and implementers, and perhaps holding of strong beliefs and negative attitudes towards vaccination by some parents or caretakers. This finding is in line with a previous study by Allison Portnoy and colleagues on the impact of SIAs on reaching children missed by routine programs, which found that the proportion of zero-dose children reached by SIAs ranged from a low 28% in Sao Tome and Principe to a high 91% in Nigeria [4]. Similar findings were also corroborated by Winter and Barchi in their SIA assessment studies that revealed only 20% of children with no prior doses of measles vaccine being

reached by the SIA in Honduras, and up to 22% of zero-dose children in Indonesia were reached by the SIA [25].

Caretaker's education was important in measles vaccination during the campaign. This analysis showed that the prevalence of non-vaccination was lower among children whose parents had at least primary school education than that among children from caretakers with no formal education. Thus, educated mothers had significant chance of immunizing their children during the campaign. This finding is consistent with the findings of other multilevel analysis studies conducted in 24 Sub Saharan African countries by Wiysonge and colleagues [26], and other cross-sectional studies conducted by Abadura et al [27]. Educated mothers know the importance of immunization. And moreover, education provide greater knowledge of health care utilization and the ability to respond to new knowledge more rapidly [28].

Caretaker's occupation seemed to be important in getting children vaccinated during the campaign. This research revealed that the prevalence of non-vaccination was 28% greater among children whose caretakers were unemployed than that of their counterparts from employed caretakers. Previous studies also found occupation to be important in influencing childhood immunization [7]. On the other hand, some studies found employment to hinder a caregiver from seeking out immunization services because of lack of time, even in the context of SIAs. However, we noted that those employed caretakers who participated in this study had also acquired some education. Therefore, they were able to find some time to have their children vaccinated because they understood the importance of the vaccination campaign.

The analysis also showed that children from older caretakers had significant chance of being vaccinated against measles. The prevalence of non-vaccination among children from caretakers or mothers who were aged 25–34 years was 25% lower than that of children from caretakers who were younger than 25 years. Some authors have attributed this to experience that older mothers gained over time on the importance of immunization, and perhaps also their knowledge on fatalities that occur to children because of lack of immunization [29]. This finding is in line with that which Babirye and colleagues also found [30].Sridhar and colleagues also reported maternal age as one of the determinants of vaccination [7].

We also found that birth order was an important factor in this research, showing a protective effect against non-vaccination. Younger children had greater chance of being vaccinated against measles during the SIA, thus. the prevalence of non-vaccination was lower among children with higher birth order. This sounds perversely counter-intuitive and not consistent with the findings of other studies [8, 9]. The likely reason for this finding could be that some parents did not perceive any measles threat to their older f children, like first and second born, who most of them were 3 to 5 years old during the time the vaccination campaign was implemented. Therefore, they did not participate in the measles SIA. The prevalence of non-vaccination was lower among children with history of previous measles vaccination than those children with no history of previous vaccination. This finding concurs with what Allison Portnoy and colleagues also found in their study about assessing the impact of SIAs [4].

The key informant interview findings are to a large extent complementing and corroborating our survey findings. Messages and information about the SIA were not well disseminated and explained to the households. Some mothers did not participate in the SIA with their eligible children because information about the SIA did not reach them. Some parents perceived that younger children only get vaccinated routinely at health facilities or at under-five clinics. Therefore, they did not see it as a need to have their children vaccinated during the campaign. This is an indication that health education and communication was not carried out effectively to the masses. Effective health education and health education have positive impact on health services utilization [31].

Flaws in the health information system were evident with the use of wrong population estimates that negatively affected the implementation of the SIA. According to the World Health Organization, sound and reliable information is the foundation of decision-making across all health system building blocks. Accurate information is essential for health system policy development and implementation, governance and regulation, health research, human resources, health education and training, service delivery and financing. It was evident in Lilongwe district that problems in health information system and health education delivery had direct negative impact in service delivery during the implementation of the 2017 measles SIA. Thus, use of wrong population estimates in some health sub districts (HSD) led to logistical problems and allocation of fewer health care workers than required. A study conducted in Kenya by Kinara also highlighted cracks in health information management system which affect health planning and service delivery because of availability of limited data, and of poor quality [32].

## Study limitations

Given that the information on measles immunization was recorded retrospectively using immunization card or maternal self-reports where the card was unavailable, mothers might not have been in a position to recall very well all the past immunization events. Therefore, respondents were prone to recall error and perhaps forgetting. The WHO and some authors argue that recall by vaccination card is considered the best practice for determining vaccination coverage in a household survey and is preferable over self-reported recall. However, some previous studies also concluded that maternal self-reports are as good as vaccination cards [20, 21]. And the interview questions were created in such a way that such recall errors should be reduced. Another limitation was that we were unable to reach out to some key stakeholders involved in the immunization programme such as medical officers, district health officers and cold chain technicians. Since the selection of households within the clusters was done purposively, the study might have suffered from selection bias. And finally, as with all cross-sectional studies, we can only describe the associations between the outcome and potential determinants; we cannot infer causality.

## Conclusions

Many children (41%) were left unvaccinated during the campaign. There was little impact of the SIA on reaching children missed by routine services as more than 50% of the children who did not receive measles vaccine under routine immunization program also missed the opportunity to be vaccinated against measles during the SIA. The District Health Team of Lilongwe district should endeavor to implement high quality supplementary immunization activities (SIAs) to reach all children, including those missed by routine immunization program. This can be achieved by involving health workers from health sub districts in planning because they know the sizes of their catchment areas by actual head counts other than using some population estimates. There was a positive impact of caretaker's education level on vaccination of eligible children. Additionally, children with low birth order had significantly lower chances of being vaccinated during the mass vaccination campaign.

Efforts are needed to enhance formal education among the communities with gender parity at the fore front. The DHT should educate the caretakers on the importance of child immunization, thereby vaccinating children irrespective of birth order. In addition, there is need to intensify community mobilization as part of SIA activities. And finally, there was poor delivery of health education to the communities; and flaws in health information system led to logistical challenges and allocation of few health care workers in some areas, which ultimately impacted on the service delivery during the mass vaccination campaign. Media outreach should be

increased among the population, and government can use this channel to disseminate standard information about any preventive health program including supplementary immunization activities.

## Supporting information

**S1 File. Quantitative questionnaire.** An original interviewer-administered questionnaire which was later transferred to Open Data Kit (ODK) to be ectronically administered. These questions were developed specifically for this study.
(DOCX)

**S2 File. Key informant guide.** This guide was used to collect qualitative data through Key Informant Interviews.
(DOCX)

## Acknowledgments

We thank the study participants, community leaders, the research assistants and the coordinator for the Expanded Programme on Immunization in Lilongwe district for their support.

## Author Contributions

**Conceptualization:** Hamilton Wales Kainga, Steven Ssendagire, Fred Wabwire Mangen.

**Data curation:** Hamilton Wales Kainga.

**Formal analysis:** Hamilton Wales Kainga, Steven Ssendagire, Fred Wabwire Mangen.

**Funding acquisition:** Hamilton Wales Kainga.

**Investigation:** Hamilton Wales Kainga.

**Methodology:** Hamilton Wales Kainga, Steven Ssendagire, Jacquellyn Nambi Ssanyu, Fred Wabwire Mangen.

**Project administration:** Hamilton Wales Kainga.

**Resources:** Hamilton Wales Kainga.

**Software:** Hamilton Wales Kainga.

**Supervision:** Steven Ssendagire, Fred Wabwire Mangen.

**Validation:** Hamilton Wales Kainga, Fred Wabwire Mangen.

**Visualization:** Fred Wabwire Mangen.

**Writing – original draft:** Hamilton Wales Kainga, Steven Ssendagire, Fred Wabwire Mangen.

**Writing – review & editing:** Hamilton Wales Kainga, Jacquellyn Nambi Ssanyu, Sarah Nabukeera, Noel Namuhani, Fred Wabwire Mangen.

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
