## [Decision Letter · Decision Letter 0]

4 Aug 2020

PONE-D-20-10234

Proportion of children aged 9 – 59 months reached by the 2017 measles Supplementary Immunization Activity among the children with or without history of measles vaccination in Lilongwe district, Malawi.

PLOS ONE

Dear Dr. Kainga,

Thank you for submitting your manuscript to PLOS ONE. After careful consideration, we feel that it has merit but does not fully meet PLOS ONE’s publication criteria as it currently stands. Therefore, we invite you to submit a revised version of the manuscript that addresses the points raised during the review process.

We look forward to receiving your revised manuscript.

Kind regards,

Kavita Singh Ongechi

Academic Editor

PLOS ONE

Journal Requirements:

3. When reporting the results of qualitative research, we suggest consulting the COREQ guidelines: http://intqhc.oxfordjournals.org/content/19/6/349. In this case, please consider including more information on the number of interviewers, their training and characteristics; and how participants were recruited.

4. Please ensure that you refer to Figures 1-3 in your text as, if accepted, production will need this reference to link the reader to the figure.

5. We note you have included a table to which you do not refer in the text of your manuscript. Please ensure that you refer to Table 2 in your text; if accepted, production will need this reference to link the reader to the Table.

6. Your ethics statement must appear in the Methods section of your manuscript. If your ethics statement is written in any section besides the Methods, please move it to the Methods section and delete it from any other section. Please also ensure that your ethics statement is included in your manuscript, as the ethics section of your online submission will not be published alongside your manuscript.

Additional Editor Comments (if provided):

Dear Author,

Please see the reviewer comments and in addition I have a few comments.

1) Introduction: Explain more about there is measles resurgence every three years.

2) Methods: For some variables consider collapsing categories because of small cell size. For marital status consider categories of currently married and currently not married. For employment, what is the difference between housewife and unemployed. Could these categories be combined.

Sincerely,

Kavita Singh Ongechi

Reviewers' comments:

Reviewer's Responses to Questions

**Comments to the Author**

1. Is the manuscript technically sound, and do the data support the conclusions?

Reviewer #1: Yes

Reviewer #2: Partly

2. Has the statistical analysis been performed appropriately and rigorously? 

Reviewer #1: Yes

Reviewer #2: I Don't Know

3. Have the authors made all data underlying the findings in their manuscript fully available?

Reviewer #1: Yes

Reviewer #2: No

4. Is the manuscript presented in an intelligible fashion and written in standard English?

Reviewer #1: Yes

Reviewer #2: Yes

5. Review Comments to the Author

Reviewer #1: This is an interesting study with potentially important implications for assessing the effectiveness of supplemental immunization activities. The manuscript is well written with coherent flow of information and argument. The conclusions may be of interest to countries working to reach children that are consistently missed in routine and supplemental immunization services. I have no major issues with the paper and offer the following minor recommendations for revision:

Abstract

1. On line 1, include the month along with year the SIA was conducted.

2. The second sentence where the authors state "...20 households..." is inconsistent with the methods section and should be harmonized with the description in the methods.

3. Line 4 in the methods description in abstract where the authors refer to "true description of when and where the child was vaccinated." The word true is subjective and should be deleted as the data are based on self-report, without an objective way of validating the caregiver's report.

4. Define MOV at first mention.

Background

5. Second paragraph, the WHO recommendation referenced was updated in 2017 and removed the MCV1 threshold for introducing MCV2. This statement should be edited to include the current position. See: https://www.who.int/immunization/policy/position_papers/WHO_PP_measles_vaccine_summary_2017.pdf?ua=1

Methods

6. Last statement in the description of the study sample "Geographical access to all levels of health care has remained stagnant especially among under five children living in the urban slums and outskirts of the city" is unclear and should be edited for clarity.

7. Study design: Indicate the respective targets of the quantitative and qualitative portions of the study. As written, it appears like the quantitative and qualitative data were collected from the same sample.

8. Page 7, sampling procedure: "In each selected village, 19 eligible households were selected." This is inconsistent with what is stated in the abstract where the authors state that 20 household were selected in each village. This should be checked and harmonized for consistency.

9. Page 8, training of research assistants: indicate how many research assistants were trained.

10. Page 8, data collection and measurements: "a child was taken to have been vaccinated during the measles SIA if the caretaker gave a true description of when and where the child was vaccinated." Similar to my comment on the abstract, it is unclear how the research team/enumerators validated the veracity of these reports. This is a subjective statement and the word "true" should be deleted.

11. First paragraph on page 9, a rationale for how the required number of KIIs was determined should be provided.

12. Page 9, still on data collection and measurements: "The outcome variable in this study was measles vaccination during 2017 measles SIA to an eligible child who had no any contraindication to vaccination" Is the outcome referenced here referring to the KIIs or the household questionnaire? This needs to be clarified. Also delete the word "any" from the statement.

Results - from this point forward, for some reason, all pages are numbered "1" and should be fixed.

13. Paragraph 1: "Only 16% were gainfully employed." Authors should clarify what "gainfully" means in this regard. Employed for payment in monetary terms, in-kind, self-employed, agricultural work?

14. Table 1: In caretakers educational level, "no education is repeated.

15. In the description of factors associated with MOVs and possible reasone for non-vaccination, define "APR" at first mention.

16. Table 2, add a footnote to explain what estimates are in bold font face. In addition, there has be a better way of differentiating between employment and self-employment. Consider different verbiage

Discussion

17: Generally, arguments need to be better developed in discussion section. It is not enough to state that a finding concurs with that of some other study without providing plausible explanations for the findings.

18. In second paragraph of the discussion section: "This research revealed that the prevalence of MOV was 28% greater among children whose caretakers were unemployed than that of their counterparts from employed caretakers. Previous studies also found occupation to be important in influencing childhood immunization [8]." There have been mixed findings in this regard. A plausible counter-argument is that employment may preclude a mother from taking time to seek out immunization services. In the context of SIAs which are provided free to the user, what could be the protective effect of employment? There is a need to balance the evidence in support and against.

19. "The prevalence of MOV was lower among children with higher birth order than first born children. This is not consistent with the findings of most studies. The likely reason for this finding could be that some parents did not perceive any measles threat to their first born children, who most of them were old during this time unlike other children with high birth order." The point you are trying to make here is confusing and should be clarified.

20. Paragraph 4 of the discussion section: "According to the World Health Organization, sound and reliable information is the foundation of decision-making across all health system building blocks. Accurate information is essential for health system policy development and implementation, governance and regulation, health research, human resources, health education and training, service delivery and financing." A citation should be provided.

Conclusion

21. "The magnitude of missed opportunities for measles vaccination among eligible children was very high as a greater proportion of eligible children were not vaccinated during the 2017 measles supplementary immunization activity (SIA) in Lilongwe district." I believe this is an inaccurate interpretation. From these findings, more children were vaccinated than not (77 vs. 23%) during the measles SIA, and cannot be interpreted as "greater proportion of eligible children were not vaccinated."

22. The last statement in the conclusion appear out of place or does not flow coherently. I suggest it should be moved to just before the preceding statement ("And finally...").

Reviewer #2: Summary

There is great value in exploring reasons behind missed opportunities for vaccination (MOV). Since the 1980s, the WHO has had several iterations of methodology for assessing MOV. Since MOV is defined in the context of health systems failing to vaccinate an individual eligible for a vaccine, these tools are designed in the context of facilities. Moreover, MOV and supplementary immunization activities (SIA) are separate concepts where SIAs complement routine vaccination. This makes it rather confusing that the proposed study for publication explores MOV in the context of SIA. Individuals not reached through SIAs would not be MOV. If the focus of the study is to assess why children may not have been reached by the measles SIA, this would be different than examining MOV. There are issues with both the design and analysis proposed in the paper.

Specific edits

Background

• The WHO defines missed opportunities for vaccination (MOV) as an individual making contact with the health system and not receiving a vaccination they are eligible for. Supplementary immunization activities (SIA) are intended to complement routine vaccination and are not specifically considered a WHO strategy specifically for addressing MOV. These need to be clearly defined if the paper is framed in the context of MOV and SIA.

• Revisit literature on gaps in reaching children during mass measles vaccination coverage. There is documentation on how SIAs miss marginalized populations/gaps in coverage.

• Revisit citations 4-10 – these primarily look at reasons for MOV in the context of regular vaccination not people being missed through SIA.

• The background should include more robust information on measles coverage (and source) in Malawi and Lilongwe. The paper cited, which were vaccine coverage estimates based on surveillance data, is not the most appropriate data source compared to household survey and DHIS2 in Malawi. In Malawi, a DHS was conducted in 2015/16 and a SPA was conducted in2013/14. The Malawi progress report to Gavi, which includes the measles campaign cites DHS and DHIS2 data to describe measles coverage and measles surveillance data. DHIS2 data in Lilongwe during the SIA campaign period should be cited.

• It’s especially surprising that SPA findings are omitted from the paper given SPA has indicators directly related to MOV.

Methods

• Why was measles SIA not captured on vaccination cards? Recall by vaccination card is considered the best practice for determining vaccination coverage in a household survey and is preferable over self-reported recall. The WHO MOV guidelines specifically advise to not accept verbal vaccination recall from mothers.

• Insufficient justification why only eight KIIs were conducted. Were KIIs all with public health facility providers or CHAM/NGOs as well?

• Insufficient justification on how the IDI guides for both parents and health workers deviate from MOV suggested protocol and standard question approaches with core omitted questions such as “Has this child ever been vaccinated?” Also would be important to describe the choice of choosing a prompted approach for asking “why was the child not vaccinated against measles” instead of an unprompted approach since this is a core question for the study.

• Asking about sex of household head is different than asking who makes decisions in a household (preferred approach).

• Based on question wording in the KII guide, there is a potential that respondents may have confused campaign-related factors with health system-related factors that would impede in vaccination.

Results

• It is unclear how the authors define “missed the opportunity of receiving measles vaccine.” The results should cite the total number of children considered “eligible” or in need of vaccination and clearly define the denominator used for coverage estimates.

• As described above, “MOV” should not be used to describe children that were not reached by the SIA campaign.

• The qualitative interviews could be more robustly and systematically analyzed. For example, issues with the health information system are cited in the discussion as a major finding. However, the quote used to support the health information system is about flawed use of population estimates – which is independent of the health information system. Was the health officer referring to DHIS2 or census estimates? I believe DHIS2 sits with the Ministry of Health in Malawi, not the National Statistical Office, so the respondent could have been referring to just census estimates (not the health information system).

Discussion

• As described above, “MOV” should not be used to describe children that were not reached by the SIA campaign.

6. PLOS authors have the option to publish the peer review history of their article (what does this mean?). If published, this will include your full peer review and any attached files.

Reviewer #1: No

Reviewer #2: No

---

## [Author Response · Author response to Decision Letter 0]

18 Sep 2020

Manuscript: PONE-D-20-10234

Response to editor and reviewers

We are grateful to reviewers and editor for the time and effort they dedicated to providing feedback about our manuscript titled, “Proportion of children aged 9 – 59 months reached by the 2017 measles Supplementary Immunization Activity among the children with or without history of measles vaccination in Lilongwe district, Malawi”. Please see below our point-by-point response to the reviewers’ comments and queries. The page and line numbers indicated refer to the manuscript with tracked changes.

Editor’s comments

We have made sure that our manuscript satisfies PLOS ONE’s style requirements. We have followed the formatting samples given above.

We gave this manuscript to Dr Alexander Kalimbira, head of Language Department at the University of Malawi. He copyedited the draft for language usage, spelling and grammar.

3. When reporting the results of qualitative research, we suggest consulting the COREQ guidelines: http://intqhc.oxfordjournals.org/content/19/6/349. In this case, please consider including more information on the number of interviewers, their training and characteristics; and how participants were recruited.

We appreciate your comment. To address this and other comments from reviewers, we have added a sub-section of “the research team” (lines 183 – 189, pg 9); recruitment of household survey participants (lines 160 – 161 pg 8); recruitment of KII participants (line 213, pg 10)

4. Please ensure that you refer to Figures 1-3 in your text as, if accepted, production will need this reference to link the reader to the figure.

All figures used have been referred to in the text (Line 268, pg 12; Line 339, pg 16)

5. We note you have included a table to which you do not refer in the text of your manuscript. Please ensure that you refer to Table 2 in your text; if accepted, production will need this reference to link the reader to the Table.

Table 2 has been referred to in the text (line 354, pg 18)

6. Your ethics statement must appear in the Methods section of your manuscript. If your ethics statement is written in any section besides the Methods, please move it to the Methods section and delete it from any other section. Please also ensure that your ethics statement is included in your manuscript, as the ethics section of your online submission will not be published alongside your manuscript.

Additional Editor Comments (if provided):

Dear Author,

Please see the reviewer comments and in addition I have a few comments.

1) Introduction: Explain more about there is measles resurgence every three years.

We appreciate your comment. This has been well clarified in the text (lines 100 – 101)

2) Methods: For some variables consider collapsing categories because of small cell size. For marital status consider categories of currently married and currently not married. For employment, what is the difference between housewife and unemployed. Could these categories be combined.

Variable ‘marital status’ has been collapsed into two categories only (pg 16). For employment, categories “housewife” and “unemployed” have been combined (page 16)

Reviewers' comments

Reviewer #1:

This is an interesting study with potentially important implications for assessing the effectiveness of supplemental immunization activities. The manuscript is well written with coherent flow of information and argument. The conclusions may be of interest to countries working to reach children that are consistently missed in routine and supplemental immunization services. I have no major issues with the paper and offer the following minor recommendations for revision:

Abstract

1. On line 1, include the month along with year the SIA was conducted.

The month and the year the SIA was conducted has been indicated (line 21, pg 2)

2. The second sentence where the authors state "...20 households..." is inconsistent with the methods section and should be harmonized with the description in the methods.

This has been corrected, 20 replaced with 19 (line 28, pg 2)

3. Line 4 in the methods description in abstract where the authors refer to "true description of when and where the child was vaccinated." The word true is subjective and should be deleted as the data are based on self-report, without an objective way of validating the caregiver's report.

Word “true” removed and sentence rephrased (line 30, pg 2)

4. Define MOV at first mention.

MOV as a primary outcome has been removed from the whole write up and replaced with “non-vaccination” of eligible children as primary outcome.

Background

5. Second paragraph, the WHO recommendation referenced was updated in 2017 and removed the MCV1 threshold for introducing MCV2. This statement should be edited to include the current position. See: https://www.who.int/immunization/policy/position_papers/WHO_PP_measles_vaccine_summary_2017.pdf?ua=1

We agree with the reviewer, and appreciate the resource shared. This statement has been edited to reflect the current WHO’s position (lines 63 – 67, pg 3)

Methods

6. Last statement in the description of the study sample "Geographical access to all levels of health care has remained stagnant especially among under five children living in the urban slums and outskirts of the city" is unclear and should be edited for clarity.

This statement has been written again with simple and standard English for clarity (line 140 – 142, pg 7)

7. Study design: Indicate the respective targets of the quantitative and qualitative portions of the study. As written, it appears like the quantitative and qualitative data were collected from the same sample.

Respective targets of the quantitative and qualitative portions of the study clearly defined (lines 150 – 152, pg 7) 

8. Page 7, sampling procedure: "In each selected village, 19 eligible households were selected." This is inconsistent with what is stated in the abstract where the authors state that 20 household were selected in each village. This should be checked and harmonized for consistency.

Corrected for consistency

9. Page 8, training of research assistants: indicate how many research assistants were trained.

We have added a sub section of the research team where the number and training of research assistants is indicated (line 182 – 189, pg 9)

10. Page 8, data collection and measurements: "a child was taken to have been vaccinated during the measles SIA if the caretaker gave a true description of when and where the child was vaccinated." Similar to my comment on the abstract, it is unclear how the research team/enumerators validated the veracity of these reports. This is a subjective statement and the word "true" should be deleted.

The word “true” has been removed, and statement written again (line 201 – 202, pg 9)

11. First paragraph on page 9, a rationale for how the required number of KIIs was determined should be provided.

The rationally for how the number of KIIs was arrived at has been explicitly given in the text (lines 211 – 212, pg 10)

12. Page 9, still on data collection and measurements: "The outcome variable in this study was measles vaccination during 2017 measles SIA to an eligible child who had no any contraindication to vaccination" Is the outcome referenced here referring to the KIIs or the household questionnaire? This needs to be clarified. Also delete the word "any" from the statement.

The word “any” deleted from the statement.The primary outcome for this research was non-vaccination of eligible children during the SIA. Therefore, the study identified associated factors, and reasons for non-vaccination for both household survey and KIIs

Results - from this point forward, for some reason, all pages are numbered "1" and should be fixed.

Correct page numbering has been done

13. Paragraph 1: "Only 16% were gainfully employed." Authors should clarify what "gainfully" means in this regard. Employed for payment in monetary terms, in-kind, self-employed, agricultural work?

The statement has been edited for clarity (lines 298 – 299, pg 14)

14. Table 1: In caretakers educational level, "no education is repeated.

One has been deleted 

15. In the description of factors associated with MOVs and possible reasone for non-vaccination, define "APR" at first mention.

APR defined as Adjusted Prevalence Ratio at first mention just below table 2 (line 324, pg 17)

16. Table 2, add a footnote to explain what estimates are in bold font face. In addition, there has be a better way of differentiating between employment and self-employment. Consider different verbiage

We appreciate you comment. However, footnotes are not permitted as per journal requirements (PLOS ONE).

Discussion

17: Generally, arguments need to be better developed in discussion section. It is not enough to state that a finding concurs with that of some other study without providing plausible explanations for the findings.

18. In second paragraph of the discussion section: "This research revealed that the prevalence of MOV was 28% greater among children whose caretakers were unemployed than that of their counterparts from employed caretakers. Previous studies also found occupation to be important in influencing childhood immunization [8]." There have been mixed findings in this regard. A plausible counter-argument is that employment may preclude a mother from taking time to seek out immunization services. In the context of SIAs which are provided free to the user, what could be the protective effect of employment? There is a need to balance the evidence in support and against.

We appreciate the comment and thank the reviewer for the guidance. We have given a plausible explanation to balance the evidence in support and against this finding. (lines 417 – 421, pg 21)

19. "The prevalence of MOV was lower among children with higher birth order than first born children. This is not consistent with the findings of most studies. The likely reason for this finding could be that some parents did not perceive any measles threat to their first born children, who most of them were old during this time unlike other children with high birth order." The point you are trying to make here is confusing and should be clarified.

We thank the reviewer for the comment. We also noted that this finding is counter-intuitive. However, we have explained this further for clarity and better understanding (lines 431 – 437, pg 20).

20. Paragraph 4 of the discussion section: "According to the World Health Organization, sound and reliable information is the foundation of decision-making across all health system building blocks. Accurate information is essential for health system policy development and implementation, governance and regulation, health research, human resources, health education and training, service delivery and financing." A citation should be provided.

Citation provided (line 458, pg 23)

Conclusion

21. "The magnitude of missed opportunities for measles vaccination among eligible children was very high as a greater proportion of eligible children were not vaccinated during the 2017 measles supplementary immunization activity (SIA) in Lilongwe district." I believe this is an inaccurate interpretation. From these findings, more children were vaccinated than not (77 vs. 23%) during the measles SIA, and cannot be interpreted as "greater proportion of eligible children were not vaccinated."

We have made a better conclusion this time as our focus is on non-vaccination of eligible children as primary outcome. However, we still note that many children were missed because the overall proportion left unvaccinated is slightly over 41%

22. The last statement in the conclusion appear out of place or does not flow coherently. I suggest it should be moved to just before the preceding statement ("And finally...").

We have made appropriate changes as advised by the reviewer (lines 499 – 502, pg 26 – 25)

Reviewer #2:

Summary

There is great value in exploring reasons behind missed opportunities for vaccination (MOV). Since the 1980s, the WHO has had several iterations of methodology for assessing MOV. Since MOV is defined in the context of health systems failing to vaccinate an individual eligible for a vaccine, these tools are designed in the context of facilities. Moreover, MOV and supplementary immunization activities (SIA) are separate concepts where SIAs complement routine vaccination. This makes it rather confusing that the proposed study for publication explores MOV in the context of SIA. Individuals not reached through SIAs would not be MOV. If the focus of the study is to assess why children may not have been reached by the measles SIA, this would be different than examining MOV. There are issues with both the design and analysis proposed in the paper.

Specific edits

Background

• The WHO defines missed opportunities for vaccination (MOV) as an individual making contact with the health system and not receiving a vaccination they are eligible for. Supplementary immunization activities (SIA) are intended to complement routine vaccination and are not specifically considered a WHO strategy specifically for addressing MOV. These need to be clearly defined if the paper is framed in the context of MOV and SIA.

We appreciate your comment, and to address this, we have removed MOV from the whole manuscript and replaced it with “non-vaccination” of eligible children as a primary outcome in this study.

• Revisit literature on gaps in reaching children during mass measles vaccination coverage. There is documentation on how SIAs miss marginalized populations/gaps in coverage.

Literature revisited and new information added (lines 80 – 89, pg 4 – 5)

• Revisit citations 4-10 – these primarily look at reasons for MOV in the context of regular vaccination not people being missed through SIA.

Citations 4 and 6 are specifically looking at reasons for non-vaccination in the context of SIAs. However, literature shows that associated factors for non-vaccination of eligible children under routine immunization services are the same even in the context of SIAs

• The background should include more robust information on measles coverage (and source) in Malawi and Lilongwe. The paper cited, which were vaccine coverage estimates based on surveillance data, is not the most appropriate data source compared to household survey and DHIS2 in Malawi. In Malawi, a DHS was conducted in 2015/16 and a SPA was conducted in2013/14. The Malawi progress report to Gavi, which includes the measles campaign cites DHS and DHIS2 data to describe measles coverage and measles surveillance data. DHIS2 data in Lilongwe during the SIA campaign period should be cited.

• It’s especially surprising that SPA findings are omitted from the paper given SPA has indicators directly related to MOV.

We appreciate your comment. We have added a more robust information on measles vaccination coverage in Malawi, and Lilongwe District in particular by citing from DHIS2 (lines 101 – 107, pg 5)

Methods

• Why was measles SIA not captured on vaccination cards? Recall by vaccination card is considered the best practice for determining vaccination coverage in a household survey and is preferable over self-reported recall. The WHO MOV guidelines specifically advise to not accept verbal vaccination recall from mothers.

We appreciate your comment, and we agree with the reviewer. However, we have given a plausible counter-argument guided by previous studies [1-3] to balance the evidence.

• Insufficient justification why only eight KIIs were conducted. Were KIIs all with public health facility providers or CHAM/NGOs as well?

The rationally for how the number of KIIs was arrived at has been explicitly given in the text (lines 207 – 211, pg 10)

• Insufficient justification on how the IDI guides for both parents and health workers deviate from MOV suggested protocol and standard question approaches with core omitted questions such as “Has this child ever been vaccinated?” Also would be important to describe the choice of choosing a prompted approach for asking “why was the child not vaccinated against measles” instead of an unprompted approach since this is a core question for the study.

• Asking about sex of household head is different than asking who makes decisions in a household (preferred approach).

• Based on question wording in the KII guide, there is a potential that respondents may have confused campaign-related factors with health system-related factors that would impede in vaccination.

We appreciate your comment. The questionnaire was primarily designed for this research. However, most of the questions were taken and adapted from the WHO’s reference manual for cluster surveys. The respondents understood the kind of information the interviewer sought from them because all interviews in the household survey were done in vernacular language. Nevertheless, we agree with the reviewer that some questions needed to be better rephrased.

Results

• It is unclear how the authors define “missed the opportunity of receiving measles vaccine.” The results should cite the total number of children considered “eligible” or in need of vaccination and clearly define the denominator used for coverage estimates.

• As described above, “MOV” should not be used to describe children that were not reached by the SIA campaign.

Number of eligible children stated (line 293, pg 14). And this number was used as a denominator for proportion estimates. Once again, MOV has been removed and replaced with non-vaccination of eligible children as a primary outcome in the context of SIA.

• The qualitative interviews could be more robustly and systematically analyzed. For example, issues with the health information system are cited in the discussion as a major finding. However, the quote used to support the health information system is about flawed use of population estimates – which is independent of the health information system. Was the health officer referring to DHIS2 or census estimates? I believe DHIS2 sits with the Ministry of Health in Malawi, not the National Statistical Office, so the respondent could have been referring to just census estimates (not the health information system).

We appreciate your comment, and agree with you. We have therefore edited the statement for clarity (lines 369 - 370, pg 19; lines 450 – 454, pg 22 - 23)

Discussion

• As described above, “MOV” should not be used to describe children that were not reached by the SIA campaign.

“MOV” removed and replaced with “non-vaccination” as a primary outcome

6. PLOS authors have the option to publish the peer review history of their article (what does this mean?). If published, this will include your full peer review and any attached files.

Do you want your identity to be public for this peer review? For information about this choice, including consent withdrawal, please see our Privacy Policy.

Reviewer #1: No

Reviewer #2: No

1. Gareaballah, E. and B. Loevinsohn, The accuracy of mother's reports about their children's vaccination status. Bulletin of the World Health Organization, 1989. 67(6): p. 669.

2. Langsten, R. and K. Hill, The accuracy of mothers' reports of child vaccination: evidence from rural Egypt. Social science & medicine, 1998. 46(9): p. 1205-1212.

3. Valadez, J.J. and L.H. Weld, Maternal recall error of child vaccination status in a developing nation. American journal of public health, 1992. 82(1): p. 120-122.

---

## [Editor Report · Decision Letter 1]

5 Oct 2020

PONE-D-20-10234R1

Proportion of children aged 9 – 59 months reached by the 2017 measles Supplementary Immunization Activity among the children with or without history of measles vaccination in Lilongwe district, Malawi.

PLOS ONE

Dear Dr. Kainga,

Thank you for submitting your manuscript to PLOS ONE. After careful consideration, we feel that it has merit but does not fully meet PLOS ONE’s publication criteria as it currently stands. Therefore, we invite you to submit a revised version of the manuscript that addresses the points raised during the review process.

Please submit your Rebuttal Letter (Sometimes called a Response to Comments), a point by point explanation of how you responded to each of the reviewers' comments. (This had not been submitted with the revised paper.)  Also you inquired about a fee waiver. As an author from Uganda, you are eligible to apply for the PLOS Global Participation Initiative. Please do apply as you resubmit your paper with the response to comments. 

We look forward to receiving your revised manuscript.

Kind regards,

Kavita Singh Ongechi

Academic Editor

PLOS ONE

---

## [Author Response · Author response to Decision Letter 1]

17 Oct 2020

1. Figure files have been uploaded to the PACE digital diagnostic tool.

2. Rebuttal letter that responds to each point raised by the academic editor and reviewer(s) has been uploaded aa a separate file.

3. PLOS GPI application letter has been uploaded as a separate file

4. A marked-up copy of the manuscript that highlights changes made to the original version has been uploaded.

5. n unmarked version of the revised paper without tracked changes has been uploaded.

---

## [Decision Letter · Decision Letter 2]

17 Nov 2020

Proportion of children aged 9 – 59 months reached by the 2017 measles Supplementary Immunization Activity among the children with or without history of measles vaccination in Lilongwe district, Malawi.

PONE-D-20-10234R2

Dear Dr. Kainga,

We’re pleased to inform you that your manuscript has been judged scientifically suitable for publication and will be formally accepted for publication once it meets all outstanding technical requirements.

Kind regards,

Kavita Singh Ongechi

Academic Editor

PLOS ONE

Additional Editor Comments (optional):

Reviewers' comments:

Reviewer's Responses to Questions

**Comments to the Author**

1. If the authors have adequately addressed your comments raised in a previous round of review and you feel that this manuscript is now acceptable for publication, you may indicate that here to bypass the “Comments to the Author” section, enter your conflict of interest statement in the “Confidential to Editor” section, and submit your "Accept" recommendation.

Reviewer #1: All comments have been addressed

2. Is the manuscript technically sound, and do the data support the conclusions?

Reviewer #1: Yes

3. Has the statistical analysis been performed appropriately and rigorously? 

Reviewer #1: Yes

4. Have the authors made all data underlying the findings in their manuscript fully available?

Reviewer #1: Yes

5. Is the manuscript presented in an intelligible fashion and written in standard English?

Reviewer #1: Yes

6. Review Comments to the Author

Reviewer #1: (No Response)

7. PLOS authors have the option to publish the peer review history of their article (what does this mean?). If published, this will include your full peer review and any attached files.

Reviewer #1: No

---

## [Editor Report · Acceptance letter]

25 Nov 2020

PONE-D-20-10234R2 

Proportion of children aged 9 – 59 months reached by the 2017 measles Supplementary Immunization Activity among the children with or without history of measles vaccination in Lilongwe district, Malawi. 

Dear Dr. Kainga:

I'm pleased to inform you that your manuscript has been deemed suitable for publication in PLOS ONE. Congratulations! Your manuscript is now with our production department. 

Kind regards, 

on behalf of

Dr. Kavita Singh Ongechi 

Academic Editor

PLOS ONE